# Allogeneic Stem Cell Transplantation in Mantle Cell Lymphoma in the Era of New Drugs and CAR-T Cell Therapy

**DOI:** 10.3390/cancers13020291

**Published:** 2021-01-14

**Authors:** Miriam Marangon, Carlo Visco, Anna Maria Barbui, Annalisa Chiappella, Alberto Fabbri, Simone Ferrero, Sara Galimberti, Stefano Luminari, Gerardo Musuraca, Alessandro Re, Vittorio Ruggero Zilioli, Marco Ladetto

**Affiliations:** 1Department of Hematology, Azienda Sanitaria Universitaria Giuliano Isontina, 34129 Trieste, Italy; miriam.marangon@asfo.sanita.fvg.it; 2Section of Hematology, Department of Medicine, University of Verona, 37134 Verona, Italy; carlo.visco@univr.it; 3Hematology Unit, ASST Papa Giovanni XXIII, 24127 Bergamo, Italy; abarbui@hpg23.it; 4Division of Hematology, Fondazione IRCCS, Istituto Nazionale dei Tumori, 20133 Milan, Italy; annalisa.chiappella@istitutotumori.mi.it; 5Hematology Division, Department of Oncology, Azienda Ospedaliero-Universitaria Senese, 53100 Siena, Italy; a.fabbri@ao-siena.toscana.it; 6Hematology Division, Department of Molecular Biotechnologies and Health Sciences, Università di Torino, 10126 Torino, Italy; simone.ferrero@unito.it; 7Hematology 1, AOU Città della Salute e della Scienza di Torino, 10126 Torino, Italy; 8Hematology Unit, Department of Clinical and Experimental Medicine, University of Pisa, 56126 Pisa, Italy; sara.galimberti@med.unipi.it; 9Hematology Unit, Azienda Unità Sanitaria Locale IRCCS di Reggio Emilia, 42123 Modena, Italy; stefano.luminari@unimore.it; 10Surgical, Medical and Dental Department of Morphological Sciences Related to Transplant, Oncology and Regenerative Medicine, University of Modena and Reggio Emilia, 42123 Modena, Italy; 11Department of Hematology, IRCCS—Istituto Scientifico Romagnolo per lo Studio e la Cura dei Tumori (I.R.S.T.), 47014 Meldola, Italy; gerardo.musuraca@irst.emr.it; 12Hematology Unit, ASST Spedali Civili, 25123 Brescia, Italy; alessandro.re@asst-spedalicivili.it; 13Division of Hematology, ASST Grande Ospedale Metropolitano Niguarda, 20162 Milan, Italy; vittorioruggero.zilioli@ospedaleniguarda.it; 14SC Ematologia, Azienda Ospedaliera Santi Antonio e Biagio e Cesare Arrigo, 15121 Alessandria, Italy; 15Dipartimento di Medicina Traslazionale, Università del Piemonte Orientale, 15121 Alessandria, Italy

**Keywords:** mantle cell lymphoma, allogeneic stem cell transplantation, Car-T cell therapy

## Abstract

**Simple Summary:**

Mantle Cell Lymphoma (MCL) is a lymphoproliferative disorder which represents less than 10% of all non-Hodgkin Lymphomas. The typical course of MCL is characterized by several relapses (“remitting-relapsing” course), and since its identification it has been considered an incurable disease. Allogeneic stem cell transplantation (allo-SCT) has represented in the past years the only treatment which could ensure prolonged remissions, at least in younger patients. In our paper, we critically revised the available data on the use of allo-SCT in MCL. The aim of our review is to identify the subgroups of patients who could best benefit from this therapeutic strategy, the optimal timing for transplantation and the best ways to bridge patients to allo-SCT, in an era in which many novel agents have been developed.

**Abstract:**

MCL is an uncommon lymphoproliferative disorder that has been regarded as incurable since its identification as a distinct entity. Allogeneic transplantation for two decades has represented the only option capable of ensuring prolonged remissions and possibly cure. Despite its efficacy, its application has been limited by feasibility limitations and substantial toxicity, particularly in elderly patients. Nevertheless, the experience accumulated over time has been wide though often scattered among retrospective and small prospective studies. In this review, we aimed at critically revise and discuss available evidence on allogeneic transplantation in MCL, trying to put available evidence into the 2020 perspective, characterized by unprecedented development of novel promising therapeutic agents and regimens.

## 1. Introduction

Mantle cell lymphoma (MCL) is an uncommon lymphoproliferative disorder accounting for 6–7% of non-Hodgkin lymphomas and is characterized by the translocation of the Cyclin D1 gene [1]. This lymphoma subtype is more frequent among males and is typical of advanced age with a peak incidence in the seventh decade. MCL is considered an incurable disease showing a typical relapsing remitting clinical course and a median survival time of approximately 5 years. The Mantle cell International Prognostic index (MIPI) can be used to identify three risk groups with a different five-year overall survival (OS) of 60%, 35%, and 20%, for the low, intermediate and high risk group, respectively; additionally, high Ki67 proliferation index and TP53 mutations are adverse prognostic findings and could imply the need for adapted treatment [2,3,4]. Survival of MCL has evolved over the past decades since immunochemotherapy regimens became available. The choice of optimal treatment is based on the aggressiveness of the disease, and on patient features (including age, performance status, and comorbidities). Treatment can be postponed in patients without symptoms, with leukemic non-nodal disease that usually carry hypermutated Immunoglobulin heavy chain (IGH) genes, with SOX11 negative disease, and without a complex karyotype.

For young and fit patients, high dose cytarabine (Ara-C) containing programs followed by ASCT are recommended [5,6]. Maintenance with rituximab has been shown to prolong PFS and OS [6]. Furthermore, maintenance with Lenalidomide showed a PFS advantage [7]. Other intensive therapies with rituximab + hyperfractionated cyclophosphamide, vincristine, doxorubicin and dexamethasone (R-hyperCVAD) alternating with HD ARA-C and methotrexate (MTX) regimen [8,9] can be considered and allow for sparing ASCT, but carry a treatment related mortality which is not inferior to that of ASCT-based programs. Less aggressive therapies for elderly patients include bortezomib + rituximab + cyclophosphamide + doxorubicin and prednisone (VR-CAP), bendamustine + rituximab (BR), and rituximab + bendamustine + cytarabine (R-BAC) regimens [10,11,12,13,14]. The use of maintenance therapy with rituximab has shown to improve patients’outcome after R-CHOP [10]. For patients who experience relapse or progression, second line regimens include non-cross resistant immunochemotherapy or new agents with documented activity. Allogeneic stem cell transplant (allo-SCT) represents a potentially curative option for younger patients. In this review article we provide an update of main available data related to the use of allo-SCT in MCL, and critically review the use of allo-SCT in the era of novel agents and Chimeric Antigen Receptor T (CAR-T) cell therapies.

## 2. Materials and Methods

This review was conducted by a panel of senior authors with expertise in research and clinical practice in MCL. In an initial meeting, which was held in Milan on 17 February 2020, the panel identified the areas of major concern and unmet needs regarding the use of allo-SCT in MCL. The main themes that emerged were indications to allo–SCT, bridging therapy, conditioning regimens, maintenance and salvage treatments for relapses and a comparison between allo–SCT and CAR-T cell therapy. Then, a review of the current state of knowledge on allo-SCT in MCL was performed. Articles published since 2000 on international journals were included in this review. In a subsequent web meeting, which was held on 27 May 2020 all the main points were discussed, some contents were added, and others were reduced. A first revised draft was reviewed from all authors and re-discussed in a web meeting on 22 October 2020.

## 3. Outcomes with allo-SCT

Allo-SCT may represent a therapeutic option for about 20% of all diagnosed MCL cases [15]. Unfortunately, this procedure is characterized by roughly 20% transplant-related mortality (TRM), 40% grade 2–4 acute graft-versus-host disease (aGVHD), 30% extended chronic-graft-versus host disease (cGVHD). Nevertheless, 5-year overall survival (OS) and progression-free survival (PFS) range between 40–60% and 30–50%, respectively [16,17,18]. Most published series are retrospective, generally small and include heterogeneous groups of patients. However, these studies have provided convincing evidence of the existence of an allogeneic graft-versus-MCL effect [18,19,20] and document the achievement of long lasting remissions in a proportion of patients which are suggestive of a curative potential.

These data support the idea that allo-SCT should be considered a well-weighed choice in the clinical practice and each physician should consider several aspects before suggesting this strategy. In the following part of the manuscript, we will analyze the position of allo-SCT within the complex MCL therapeutic scenario, with special attention to the profile of the “very high-risk” younger patients, where allo-SCT appears particularly indicated.

## 4. Which Is the Best Time for allo-SCT?

One of the most relevant objects of debate is if allo-SCT might be positioned as consolidation of second-line therapy or if it could be adopted upfront in very selected cases (Table 1).

In the ESMO (European Society for Medical Oncology) guidelines edited in 2017, the authors stated that there were no data enough to support the application of allo-SCT as part of front-line treatment [level of evidence: II, grade of recommendation: D]. However, allo-SCT, appeared a potentially curative option in the subgroup of early relapsed/refractory younger patients [level of evidence: III, grade of recommendation: B] [21]. The Japanese guidelines position allo-SCT in first relapse, after ASCT, but only for patients aged less than 61 years [22]. Similarly, the British guidelines consider allo-SCT in first relapse, preferably after a Bruton’s Tyrosine Kinase inhibitor (iBTK), to increase the number of cases in good response at transplant [23]. During the European Society for Bone Marrow & Transplantation (EBMT) meeting held in 2019, experts strongly sustained that a second ASCT is not an appropriate option in MCL patients failing the first autologous procedure, while allo-SCT should be seriously considered in these cases [24] The same indication (young relapsed/refractory patients) has been also fully accepted in the 2020 version of the NCCN guidelines: allo-SCT is indicated as consolidation after a second-line treatment [25].

The updated version of the European guidelines confirmed that allo-SCT may represent the best option for young and fit subjects who relapse within 24 months from the first line treatment, but it also suggests to consider allo-SCT for cases with later relapse [26].

A threshold of 24 months has been described to dichotomize patients with meaningful better (progression of disease (POD) > 24 months) or worse outcome (early POD) [27]. In a real-life series from the Fondazione Italiana Linfomi (FIL), of 188 relapsed or refractory patients, forty-one (22%) underwent allo-SCT, of whom 28 (68%) were transplanted in second remission and 13 (32%) in third remission or later. The performance of allo-SCT had a favorable significant impact on survival of patients with early-POD, being capable of rescuing a significant fraction of these patients from early death [27].

The international retrospective “Mantle First” study, including 258 MCL patients in first relapse, evaluated outcome after second line treatment comparing ibrutinib, R-BAC, R-bendamustine or other approaches. A poorer outcome was observed in patients with early POD independently of the second line regimen; again, in this subgroup, allo-SCT conferred an advantage in terms of survival [28]. On the other hand, in a series of 360 patients, the EBMT reported that remission duration after ASCT significantly affected the outcome of salvage allo-SCT: in fact, patients who relapsed less than one year after ASCT had a poor prognosis, even if treated with allo-SCT [29].

Several studies demonstrated a similar outcome when allo-SCT was performed in the relapse setting, as compared to its use as consolidation of first line high dose chemotherapy. For example, the German group, in a series of 39 MCL patients, reported a TRM of 24%, a 5-year PFS of 67%, and an OS of 73%, without substantial difference between patients receiving transplant as consolidation of ASCT or at relapse [30]. The absence of an evident advantage deriving from up-front allo-SCT has been also stated in the EBMT position paper edited in 2015 [31]. On the other hand, the British group reported that RIC allo-SCT as first-line consolidation was associated with promising 2-year PFS and OS of 68% and 80%, respectively [32]. These percentages compare favorably with those previously reported for RIC allo-SCT at relapse, where the reported 2-year event-free-survival (EFS)/ OS were 50% and 53%, respectively [19]. Another series of 519 patients receiving allo-SCT in first partial or complete remission or at relapse was analyzed by the Center for International Blood and Marrow Transplant Research (CIBMTR). In this series, a significant difference between “early” RIC allo-SCT (patients who received allo-SCT in first CR or PR, with no more of 2 lines of previous therapy) and “late” RIC allo-SCT (patients who received allo-SCT at relapse) was observed; in fact, 5-year OS was 62% and 31%, respectively [33]. Accordingly, the British society found that having < 2 prior lines of therapy positively influenced survival [20].

On the other hand, in an EBMT study no differences were reported for patients who received allo-SCT at first chemosensitive response or later in the course of disease, regardless of the number of regimens needed to achieve this status. Finally, a French study found no survival difference according to number of lines of treatment prior to allo-SCT [17,34].

Comparison of different studies with such a great heterogeneity in terms of patients clinical and biological features may be tricky and unjustified. Furthermore, biological characterization of patients that were sent to allo-SCT is lacking.

### Key Points

Patients experiencing early relapse or chemorefractoriness, similarly to those with an unsatisfying response to the second-line regimens, should be considered candidates for allo-SCT. The majority of the studies demonstrated a comparable survival when allo-SCT was performed in first remission or within second relapse, suggesting that its optimal setting might be first or second relapse, as indicated by international guidelines.

## 5. Allo-SCT as an Option for “Very High Risk” Patients

In two particular subsets of patients, allo-SCT could be positioned earlier in the course of the disease, without a previous treatment with ASCT:
primarily chemorefractory subjects, after a reinduction therapy;“very high risk” cases [35].

Primary chemoresistance is a rare event, occurring in about 10% of cases [21].

Life expectancy of this selected population is very low.

The definition of “very high risk” is still matter of debate, because different risk scores have been developed in the last years but only few of them are really used in the clinical practice. MIPI and MIPI-c are based on clinical and histological features. The addition of the Ki-67 assessment to the classical MIPI (combined MIPI, MIPI-c) allows to classify patients in four prognostic groups with different outcomes (low, low-intermediate, high-intermediate and high risk), with median OS of 9.4, 4.9, 3.2, and 1.8 years respectively [3].

MIPI-c is commonly used in the real life because it includes easily available parameters; nevertheless, many researchers tried to increase its predictive power by exploring other biological items, such as microRNA (miRNA) expression [36] and gene expression profiling [37,38]. Biological signatures based on gene expression profiling are not easily performable, and, in our opinion, hardly to be implemented in the clinical context. On the contrary, all molecular laboratories are today equipped for the mutational screening of TP53. TP53 mutation has been reported in 62% of MCL blastoid/pleomorphic variants [39] and in 20–28% of MCL nodal type, either in clonal or subclonal form [40]. Recent studies have demonstrated the strong negative impact of TP53 mutations on outcome of MCL patients, in terms of worse EFS and OS [2,41].

The Fondazione Italiana Linfomi (FIL) MCL0208 trial found that TP53 mutations (tested by NGS) were associated with a dismal outcome: indeed, TP53-mutated patients presented lower median PFS (17 months vs. not reached) and shorter OS (51 months vs. not reached) in respect of wild-type cases. Interestingly, in the TP53-mutated subgroup, neither MRD-negativity before or after ASCT nor the addition of maintenance with lenalidomide seemed able to overcome the negative prognostic impact of TP53 mutations [42]. In addition, in this trial the association of mutations of KMT2D with poor outcomes in terms of PFS and OS was demonstrated. A new prognostic index, called “MIPI-g”, was developed by adding TP53 and KMT2D mutations to MIPI-c, thus leading to the identification of a subgroup of high risk patients who could benefit from novel therapeutic strategies and perhaps a front-line allo-SCT [43].

Allo-SCT seemed to eliminate prognostic differences related to TP53 status: in a cohort of 42 patients who underwent allo-SCT, 2-year OS was 80% and 70%, respectively, for TP53 wild-type and TP53 mutated cases, without a statistically significant difference. This is mostly due to the TP53 independent death mechanisms associated to immunological killing. One alternative hypothesis suggested by the authors is that TP53 alteration would modify the host immunocompetence inducing an immune-suppressive tumor microenvironment and that allo-SCT might have the potential to replace the dysfunctional host immune system and to synergize with chemotherapy [44].

Alternative ways to identify the “very high risk” patients to whom propose allo-SCT include MRD testing and FDG-PET. However, the presence of minimal active disease at the time of allo-SCT does not appear to preclude long-term remissions in advanced MCL patients, differently from what was observed for chemotherapy and ASCT [45]. Moreover, to date no studies have evaluated the role of pre transplant MRD positivity on allo-SCT outcome.

### Key Points

Allo-SCT represents an option always to be considered in refractory cases; moreover, emerging data show its potential curative role in a “very high risk” population. Unfortunately, a unique definition of “very high risk” is still lacking, and many of the tests necessary for calculating the biological risk are not practicable in the routine practice. We suggest that “very high risk” patients are those with at least two of the following adverse features: high proliferative index, TP53 mutations, blastoid morphology.

Figure 1 represents the positioning of allo-SCT in transplant-eligible patients:
after intensive immuno-chemotherapy (orange box), young and fit patients who achieve complete response (CR) proceed with maintenance (green arrow, green box).In case of partial response (PR), no response (SD) or relapse/progression (PD), patients receive re-induction therapy (BTK inhibitors, chemo-immunotherapy, investigational drug) (violet arrow, violet box).All patients relapsed before 24 months from the end of the first-line therapy who achieve a CR proceed immediately to allo-SCT (dark blue arrow, blue box).Subjects who relapse later than 24 months proceed to allo-SCT at the first signs of failure of second-line therapy (light blue arrow, blue box) if <60 years, with matched (related) donor.Patients who do not respond to second-line treatment, receive a third-line therapy (venetoclax, investigational drugs, chemo-immunotherapy) and proceed to allo-SCT at reaching of PR/CR (red arrow, red box, blue box).For cases at very high risk (blastoid variant, TP53-mutated), allo-SCT could be considered as first-line consolidation in some selected patients and discussed with each patient.

## 6. Chemosensitivity and the Role of allo-SCT

Even if chemosensitivity prior to allo-SCT is the stronger predictor of outcome, prolonged remissions seem to be possible even among patients with chemotherapy unresponsive MCL. Hamadani et al. identified 202 patients with chemorefractory MCL reported to CIBMTR; 128 received RIC-allo-SCT. 3-years NRM was similarly high regardless of conditioning intensity (47% for MA, 43% for RIC), suggesting that patients with active disease have significantly increased risk of transplant-related complications. However, the 3-year PFS and OS were 25% and 30%, respectively, with no difference after myeloablative or RIC conditioning [46].

On the other hand, the previously cited registry study by EBMT on 324 MCL patients receiving RIC-allo-SCT reported that the obtainment of CR or PR to the last pre transplant treatment was the best predictor of outcome and chemosensitive patients had less aGVHD. In this series, the cumulative incidence of relapse was 25% and 40% at 1 and 5 years, respectively, and patients with chemorefractory disease had a substantially higher risk of relapse and worse PFS and OS [17]. Tessoulin et al. reported a French national survey in 106 patients who failed after auto-SCT and received RIC-allo-SCT. TRM and NRM were not affected by disease status. However, median PFS and OS for patients who received the transplant at least in PR were 34 and 63 months, compared with 4 and 6 months for patients in PD [34]. The British society reported chemosensitivity as the only factor associated with reduced relapse rate in 70 patients receiving RIC-allo-SCT, with 3-year OS and PFS for patients transplanted in CR, PR, and with refractory disease of 60% and 31%, 40% and 26%, and 38% and 0%, respectively [20]. More recent data from EBMT on a small number of patients, who received Ibrutinib pre allo-SCT, confirmed that PFS was significantly worse in patients who had failed ibrutinib compared to the ibrutinib-sensitive subset [47].

Of note, in the cited studies a proportion of patients (15–25%) were unresponsive to the last pre allo-SCT line of treatment; however, also non-responding patients underwent allo-SCT.

Nowadays, FDG-PET is widely performed before allo-SCT in order to identify persistent disease. Bachanova et al. evaluated the role of PET response before allo-SCT in patients in radiologic complete or partial response (documented by CT scan) and demonstrated that, in MCL, a positive pretransplantation PET was associated with an increased risk of relapse/progression. However, pre allo-SCT PET status did not appear as a strong predictor of survival, at least in chemosensitive disease by conventional radiographic criteria, and residual PET positivity should not be interpreted as a barrier to a successful allograft [48].

## 7. Bridging Strategies

It is not easy to define which strategies might be the best way to bridge R/R MCL patients to allo-SCT. Evidences on the role of transplant in MCL patients are represented by registry studies and refer to patients treated with disparate chemotherapy approaches [33], Recent studies on new drugs rarely include patients subsequently undergoing allo-SCT as salvage treatment, being mainly PFS the primary objective of most studies. Then, we can derive indirect evidences on the possible role as bridge to transplant of the available schemes, basing on the published data in terms of ORR, CR, duration of response (DOR) and PFS.

### 7.1. Targeted Therapies

Ibrutinib is the most widely used single agent therapy for R/R MCL and it permits the achievement of an ORR of 77% (CR 23%) with a median PFS of 15.6 months. It is reported to be more efficacious as early salvage therapy with published data of 33% CR and median PFS 25.4 months when used as second-line treatment [49]. The previously cited EBMT study on 22 MCL patients treated with ibrutinib and subsequent allo-SCT demonstrated a low NRM incidence and a low rate of disease recurrence, which translated into 12-month PFS > 75% for the whole MCL sample (>90% for the ibrutinib sensitive subset). Thus, on one hand ibrutinib bridging appears to be a promising approach for improving feasibility and efficacy of allo-SCT; on the other hand, allo-SCT seems to be a reasonable consolidation strategy in patients with R/R MCL responding to ibrutinib. In this study, median exposition of patients with MCL to ibrutinib was 149 days (significantly shorter than the exposure of the CLL cohort); given the known median DOR of ibrutinib salvaged patients and the poor outcome of patients with MCL progression under ibrutinib, allo-SCT should be performed as soon as best response has been achieved [47].

Data on acalabrutinib and zanubrutinib seem to go in the same direction as ibrutinib efficacy data, but a more extensive follow-up is needed to confirm such results [50,51].

Venetoclax seems to be a promising approach, as depicted in a recent UK retrospective study; however, data are still immature (in fact, in this study only one patient underwent allo-SCT after achieving a response to venetoclax) and the study was conducted in heavily pre-treated patients, particularly iBTK-failure patients (ORR 53%, median PFS 3.2 months) [52].

Encouraging results have also been observed in R/R MCL with the combination of venetoclax and ibrutinib (71% CR, 12-months PFS 75%); therefore, these two drugs together could represent an effective bridge strategy; however, data are still immature as in the phase 2 study no patients underwent allo-SCT [53]. Further results are awaited from the phase 3 randomized trial (NCT03112174).

### 7.2. Chemo-Immunotherapy

In R/R MCL, rituximab+bendamustine has an ORR of 75% (CR 50%) and a median PFS of 18 months [54]. R-BAC has an ORR of 80% (CR 70%), with a 70% 2-year PFS [55]. In real life experiences, R-BAC has demonstrated to be a promising bridge to transplant scheme, with 22–33% of patients undergoing allo-SCT after achieving a clinical response [27,56]. Given the favorable outcomes observed in terms of ORR and CR, R-BAC can represent an alternative bridging strategy to allo-SCT. In patients who were previously exposed to bendamustine, bortezomib-based combinations (VR-CAP) may be considered as a reinduction treatment.

### 7.3. Key Points

In the era of new drugs allo-SCT still represents the only curative potential approach for patients with R/R MCL.

No systematic data are available on bridge to allo-SCT strategy. Available data indicated that ibrutinib may be the preferable bridge-to-transplant choice, while results with second generation iBTK or with the combination of ibrutinib and venetoclax are awaited.

R-BAC may represent a good option as bridge to allo-SCT in patients who have relapsed or are refractory to iBTK.

## 8. Conditioning Regimens

The choice of conditioning regimen is common to other histotypes of lymphoma, and not focused on MCL. Myeloablative, non-myeloablative and RIC-allo-SCT may be considered, but there is no clear evidence of the superiority of one of these approaches upon the others [46]. Unfortunately, there is also little evidence that any prognosticator including MRD could be of help in guiding decisions in these late treatment phases. Conditioning regimen should be therefore selected on the basis of age, hematopoietic cell transplantation comorbidity Index (HCT-CI), prior therapy, status of disease at transplant, on individual patient basis and on center experience [23].

The most used conditioning regimens for allo-SCT in Lymphoma, not only in MCL, are myeloablative conditioning (Busulfan/Cyclophosphamide, Busulfan/Fludarabine, Cyclophosphamide or Fludarabine/total body irradiation-TBI, Fludarabine/Busulfan/Thiotepa) and non-myeloablative/RIC conditioning (Cyclophosphamide/Fludarabine/TBI, Busulfan/Fludarabine).

The use of allogeneic stem cell transplantation for the treatment of MCL has been reported since the late 1990s, but the conventional myeloablative conditioning was associated with a high risk of NRM up to 40%, especially in MCL patients who are aged and often with many comorbidities [57,58], so this procedure was limited to a very small proportion of selected patients. Despite data comparing myeloablative vs. RIC-allo-SCT exclusively in younger patients do not exist, usually it is suggested to reserve myeloablative conditioning for very fit younger candidates.

Over the past two decades, nonmyeloablative and RIC regimens have been adopted; these regimens rely primarily on graft vs. lymphoma (GVL) induction, ensuring at the same time a reduced toxicity. Some data regarding RIC-allo-SCT in MCL were collected in 2018 in the previously cited study by the EBMT Lymphoma working party: a large retrospective study that included 324 patients who received RIC-allo-SCT for R/R MCL between 2000 and 2008 after Bu or Cy+ TBI+ purine analog + ATG, or melphalan + purine analog + CAMPATH using a sibling or unrelated donor [17]. Forty three percent of the patients received >3 previous lines of therapy, including ASCT in 46% of cases. Despite the advanced stage of the disease, NRM was 24% at 1 year, (mainly due to GVHD and infections) and 40% of patients relapsed/progressed by 5 years following the transplant, while relapse rate was extremely low beyond 5 years, showing that RIC-allo-SCT may be a curative procedure in MCL. One-year cGVHD rate was 41%.

Another previously cited large retrospective study evaluated RIC-allo-SCT in 106 patients who had received a prior ASCT for MCL, 72% of whom in a first line setting. NRM was described 29% after 12 months and 32% after 3 years, with a median PFS and OS of 30.1 and 62 months, respectively [34].

### Key Points

In conclusion, RIC allo-SCT has been shown to be the safest and most effective strategy; no prospective randomized trials are available to define an optimal RIC regimen among patients with NHL. Myeloablative conditioning should be reserved to selected, very “fit” younger patients.

## 9. Donor Selection and Stem-Cell Source

There is no specific recommendation for donor selection in MCL lymphoma: donor age, gender, CMV status, donor-recipient AB0 compatibility, donor-specific antibodies (DSA) in the patient, are the principal characteristics that should be evaluated. Approximately 30% of patients have an HLA matched sibling donor and this represents the established gold standard donor source. In the absence of an HLA-identical sibling, an unrelated donor (URD) who is HLA-matched to the transplant recipient at the allele level at HLA-A, -B, -C, and -DRB1 is currently considered the preferred alternative donor [59]. The likelihood of finding an HLA-matched URD varies among racial and ethnic groups, with the highest probability of success among whites of Western European descent (75%) and the lowest probability among blacks of South or Central America, at 16% [60]. Other alternative donors including haploidentical related donors or cord blood are often considered when an HLA-matched URD is not available [61,62], and have drastically increased donor availability for allo-SCT candidates.

Recently, several Asian centers have reported favorable outcomes of haploidentical transplantation (haplo-SCT), utilizing T-cell–replete grafts with intensive immunosuppression using antithymocyte globulin (ATG) [63,64].

A different strategy of T-cell–replete haploidentical transplantation being increasingly used involves administration of post transplantation cyclophosphamide, which mitigates the risk of GVHD by targeting alloreactive T cells rapidly proliferating early after an HLA-mismatched transplant, relatively sparing regulatory T cells and leaving unaffected the nondividing hematopoietic stem and progenitor cells [62].

In 2016, the CIBMTR performed a registry analysis in order to evaluate outcomes of lymphoma patients undergoing haplo-SCT and post transplantation cyclophosphamide in comparison with HLA matched sibling donors allo-SCT. Multivariate analysis showed no significant difference between haplo-SCT and HLA matched sibling donors allo-SCT in terms of NRM, Progression/relapse, PFS and OS. Three-year rates of NRM were 15 vs. 13%, relapse/progression 37 vs. 40%, PFS 48 vs. 48% and OS 61 vs. 62%. A lower risk of c GVHD was associated with RIC haplo-SCT with post transplantation cyclophosphamide at 1 year (12 vs. 45%, p ≤ 0.001) [65].

Between 2008 and 2013, the center for International Blood and Marrow Transplant Research (CIBMTR) collected data from more than 500 transplant centers worldwide, regarding 199 Hodgkin and 710 non-Hodgkin lymphoma patients (140 were affected by mantle cell lymphoma) undergoing their first RIC or NMA conditioning haploidentical or matched unrelated donor allo-SCT, these latter with ATG or without. Authors suggested that RIC haplo-SCT with post transplantation cyclophosphamide shows a similar, non-statistically different OS at 3 years (between 50–60%) when compared to matched unrelated donor allo-SCT, and that in multivariate analysis no differences were found in terms of NRM, relapse/progression and PFS in the groups. As in the previous registry analysis, a lower risk of c GVHD was associated with RIC haplo-SCT with post transplantation cyclophosphamide [62].

In 2019 Bazarbachi on behalf of LWP-EBMT tried to study if donor characteristics, stem cell source and conditioning could be associated to a different outcome in the setting of haplo-SCT with post transplantation cyclophosphamide in lymphoma. Study population were 474 adults with Hodgkin (240), peripheral T (88), DLBCL (77), mantle cell (40) and follicular lymphoma (29). No advantage of donor age on transplant outcome was found; on multivariate analysis, a decreased risk of aGVHD grade II-IV was observed when offspring donors or bone marrow cells were used. On the contrary, extensive cGVHD was higher in patients in PR at haplo-SCT or when using sisters, haploidentical donors beyond first degree, or female donor in male patients. CR at haplo-SCT improved PFS and OS, whereas these were negatively affected by CMV donor/recipient status. PFS and OS were also different in different lymphoma histotypes: in MCL patients, 2-year PFS and OS of 56% and 61% were observed, respectively [66].

A recent European retrospective study demonstrated superior outcomes in patients affected by Hodgkin and non-Hodgkin lymphomas treated with haplo-SCT in comparison with unrelated cord blood allo-SCT, in terms of OS and PFS [67].

### Key Points

Even if data were extrapolated from studies regarding all lymphoma subtypes and not only MCL, haplo-SCT seems to be non-inferior to HLA-matched URD allo-SCT and superior to unrelated cord blood allo-SCT.

## 10. Graft Versus Lymphoma

Retrospective studies consistently demonstrate significant lower relapse rate for lymphoma patients receiving allogeneic rather than autologous transplantation, even if the high TRM of allogeneic procedure tempers this advantage. The first evidence of Graft Versus lymphoma (GVL) effect was described by Jones and colleagues in 1991, and confirmed by subsequent studies [68,69,70]. Retrospective analyses demonstrated that a lower relapse rate after allogeneic transplantation was also due to the reinfusion of purged stem cells [71,72].

A GVL effect is mostly demonstrated by the durable resolution of residual or progressive disease after allo-SCT with the withdrawal of immunosuppression and the administration of donor lymphocytes infusions and by the observation that T cell depletion is associated with an increased risk of relapse, even if this is particularly true in the setting of indolent lymphoma [73,74,75,76]. Notably, no studies specifically demonstrated the presence of a GVL effect in the subset of MCL.

### Key Points

Studies conducted in different lymphoma histotypes concluded that the superiority of allo-SCT on ASCT in terms of lower relapse rates is due to GVL effect and in part due to the reinfusion of purged stem cells. No data are available regarding the evidence of a clear GVL effect in MCL; however, evidences which were demonstrated in other lymphoma subtypes could also be extended to MCL.

## 11. Maintenance

There are no systematic available data on post allo-SCT maintenance in MCL and no indications in international guidelines.

In a pilot study, rituximab has been used in combination with donor-lymphocyte infusion (DLI) as prophylaxis of relapse after allo-SCT (5 patients with MCL were included: 3 were alive and maintained a CR at last follow up, one relapsed and one died while in CR) [77]. Currently, there is one recruiting prospective phase I clinical trial which investigates administration of idelalisib as post allo- SCT maintenance in patients with B cell lymphoma, including MCL (NCT03151057).

## 12. Pre-Emptive Treatment

The role of rituximab in pre-emptive treatment of MCL patients with molecular relapse/persistence (minimal residual disease positivity, MRD+) after autologous stem cell transplantation (ASCT) is well known [78,79]. Therefore, this treatment could also be applied in the setting of MRD+ after allo-SCT.

ESMO Clinical Practice Guidelines for MCL suggest the use of DLI in MRD+ patients post allo-SCT [21].

## 13. Post allo-SCT Salvage Treatment

Most of the studies regarding salvage treatments are retrospective and conducted on few patients. Therapeutic strategies which were adopted in these series include DLI, second allo-SCT, rituximab (alone or in combination) [20,80,81].

The role of ibrutinib as post allo-SCT salvage treatment has been investigated in some studies. The previously cited EBMT study examined the outcome of allo-SCT in chronic lymphocytic leukemia (CLL) or MCL after prior exposure to ibrutinib. Interestingly, in 3 patients with MCL who had post-transplant disease recurrence, one patient was not retreated and rapidly died, another one was re-exposed to ibrutinib with transient response but died of progressive disease 8 months after relapse, and the third patient experienced MCL clearance after DLI [47].

A Case report described complete remission in a MCL patient with a CNS relapse 5 months after allo-SCT, who received ibrutinib as salvage post-transplantation monotherapy [82].

Moreover, two retrospective studies were conducted in MCL and CLL patients on the use of ibrutinib to treat post allo-SCT relapse: in the American study, eighteen-month OS and PFS rates for MCL patients (n = 6) were both 33% [83]; in the European one, 2 out of 3 patients with MCL who were treated with ibrutinib for post allo-SCT relapse obtained a CR [84].

Another EBMT study including 56 patients, confirmed that ibrutinib is an effective and safe salvage therapy after allo-SCT in CLL: this safety message might be applicable to MCL patients, too [85].

An ongoing prospective phase II clinical trial (NCT02869633) is evaluating the role of ibrutinib in patients with relapsed/refractory lymphoma (including MCL) post allo-SCT, and Other recruiting trials include the combination of selinexor and ibrutinib for the treatment of patients with relapsed/refractory CLL or Aggressive NHL, including patients who have undergone allo-SCT (NCT02303392).

A recruiting prospective clinical trial (NCT01087294) investigates the use of anti-CD19-CAR-transduced T cells infusion in patients with persisting B-cell malignancies despite allo-SCT and at least 1 standard DLI. At the moment, results on 4 patients with MCL show that three patients maintained a stable disease and one patient achieved a partial response persisting at 3 months from CAR-T cell infusion [86]. Further studies are required to better define the role of CAR-T treatment after allo-SCT.

### Key Points

No systematic available data on post allo-SCT maintenance in MCL;ESMO Clinical Practice Guidelines for MCL do not recommend MRD guided treatment in MCL, with the exception of the setting of DLI treatment post allo-SCT;salvage treatment: better considering DLI than 2 allo-SCT;salvage treatment: case reports of activity and safety of ibrutinib;a few prospective ongoing clinical trials (Table 2).

## 14. Positioning CAR-T in the allo-SCT Era, and Viceversa

MCL relapsed/refractory patients after salvage treatment with iBTK have a dismal prognosis. In this setting of patients, allo-SCT represents an option only in patients achieving a second response with salvage regimens, and the NRM, even with reduced intensity therapy, is roughly 10–24%.

The anti-CD19 CAR-T therapies are active in relapsed/refractory diffuse large B cell and aggressive lymphomas, where they represent the best third line treatment in young patients not eligible to transplant due to active disease.

Preliminary results of CAR-T in MCL showed a promising activity and prompted the conduction of a phase II study.

Wang et al. [87] recently published the final results of ZUMA-2 trial, a single arm, multicenter, phase 2 open-label trial, to evaluate the CAR-T KTE-X19 in patients with relapsed or refractory MCL. 74 patients were enrolled and 68 (92%) were infused. All patients were exposed and relapsed/refractory to iBTK and the majority of them were at high risk, with Ki67 proliferation index upper to 30% in 82%, TP53 mutation in 17% and blastoid or pleomorphic aspect in 31% of the cases.

The objective response in all patients was 85%, with 59% complete remission rate; estimated PFS and OS at 1 year were 61% and 83%. In a subgroup analysis, the efficacy of KTE-X19 was independent by high risk features. 24% of patients died: 21% due to progressive disease, while 3% died for infective complications during lymphodepleting chemotherapy before CAR-T infusion.

All patients experienced at least one adverse event, and the most frequent were hematological toxicities. Cytokine release syndrome (CRS) occurred in 91% of patients and neurological toxicities in 63% of patients; of note, no deaths related to CRS or neurologic events were observed.

The results of ZUMA-2 were impressive and may change the clinical practice in the next future.

However, there are some issues to be considered (see Table 3):
CAR-T KTE-X19 treatment is feasible and effective also in patients with active disease; allo-SCT is a curative option only in patients achieving a second response with salvage regimens.The follow-up for CAR-T KTE-X19 is short compared to allo-SCT; moreover, even if the experience of aggressive B-cell lymphomas showed that the PFS and OS rates were maintained at longer follow-up periods in patients treated with CAR-T cells, this has to be proven in MCL, a disease characterized by a continuous pattern of relapse even after many years of ongoing complete remission.The results obtained with CAR-T KTE-X19 were observed in a population with a poor prognosis, and all of the patients were pre-treated with iBTK.At the time of relapse after CAR-T KTE-X19, patients could perform a salvage treatment and could receive an allo-SCT as consolidation. On the other hand, previous allo-SCT might impact on the feasibility of a subsequent treatment with CAR-T KTE-X19. The use of allo-SCT as consolidation after the obtainment of a CR with CAR-T KTE-X19 is purely investigational.The NRM ranged between 10 to 24% with transplant strategies; NRM related to CAR-T KTE-X19 is 3%.Regarding feasibility, transplant is based on the availability of a donor (easier with the introduction of haploidentical donor). The rate of failure in the manufacture of CAR-T KTE-X19 cells was 4%. However, the relevant economic impact of CAR-T KTE-X19 cells should be considered, as well as its accessibility, which is still limited.

The integration of CAR-T into the MCL algorithm is still far from being fully established and a number of issues still need to be addressed. One of the most critical is the potential combination of CAR-T and allo-SCT as already performed in other neoplasms [88,89,90]. The long-term follow-up data following CAR-T KTE-X19 will be critical to define if CAR-T would need further consolidation or would be adequate as a “stand alone” approach. MRD evaluation could potentially also play a role in future patient selection. Future studies will help clarifying these issues.

### Key Points

CAR-T KTE-X19 therapy represents an effective and promising salvage treatment strategy in MCL with a high rate of complete responses and a low incidence of relevant toxicities; however, at the moment, follow up is too short to compare its long-term efficacy with that of allo-SCT and its feasibility is still limited.

## 15. Conclusions

For many years, allo-SCT has represented the only potentially curative option in MCL. Despite this clear clinical effect, its broad usage has always been limited by toxicity and age-related limitations. Consequently, allo-SCT has remained a “niche” treatment reserved to small highly selected patient populations. In future years, the use of allo-SCT will possibly further reduce its therapeutic window due to the advent of several new immunotherapeutic approaches. This is to the case of the major success of current treatment strategies which is not limited to the exciting achievements of cellular therapy, but encompasses better induction treatments based on biological agents and effective maintenance strategies. Nevertheless, the potential benefit of combining allo-SCT and novel immunotherapeutic approaches is also a potential field of interest particularly for high-risk patients. From a historical perspective, it should be noted that many of the treatments currently under study (cellular therapies and T-cell engaging bispecific antibodies) have been conceived based on the bulk of knowledge accumulated in the context of allo-SCT strategies. In such a perspective, novel immunotherapeutic regimens may not be considered as an alternative to allo-SCT, but rather as the ultimate and sophisticated refinement of the old concept of “immunotherapeutic effect” of allogeneic transplantation, an idea that was originally postulated more than sixty years ago [91].

## Figures and Tables

**Figure 1 cancers-13-00291-f001:**
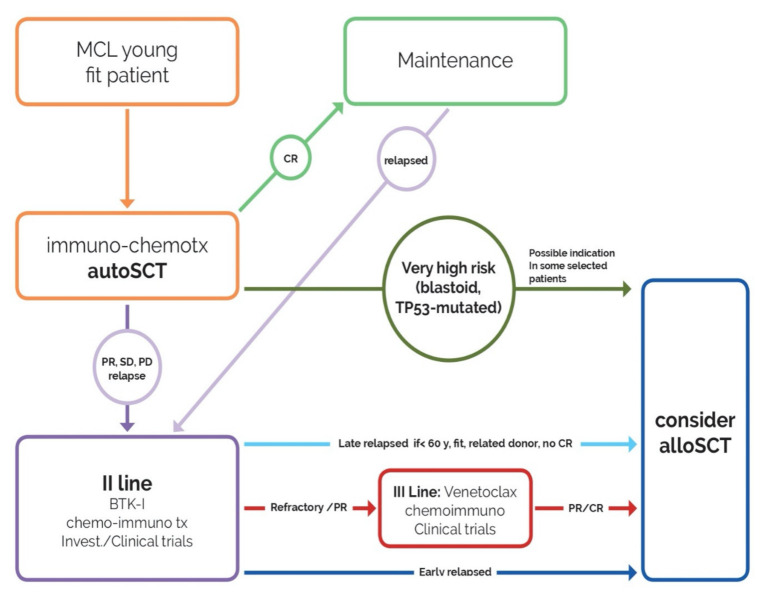
Therapeutic strategies in Mantle Cell Lymphoma (MCL) and positioning of allo-SCT.

**Table 1 cancers-13-00291-t001:** Comparison between studies showing allo-SCT as consolidation of the first or subsequent lines of treatment.

Study	Timing	N° Patients	ORR/CR	Median FU	2-yr NRM	5-yr PFS	5-yr OS	Main Toxicities
Le Gouill et al.	Salvage	70	95%/89%	24mo	32%	≈25%	≈25%	GVHD
Krüger et al.	First line	24	86%/76%	2.8yr		≈67%	≈73%	Stomatitis, infection, GVHD
	Salvage	15	91%/83%	2.8yr		≈67%	≈73%	
Rule et al.	First line	25	92%/60%	60.5mo	13%	56%	76%	Infection, mucositis, GVHD
Fenske et al.	First response	50		48mo	≈30%	55%	62%	
	Salvage	88		37mo	≈25%	24%	31%	
Tessoulin et al.	Salvage	106	97%/86%	45mo	≈30%	≈35%	≈55%	GVHD, infection

[19,30,32,33,34].

**Table 2 cancers-13-00291-t002:** Ongoing clinical trials enrolling patients with MCL after allogeneic transplantation.

Treatment Drugs	Trial Phase	Recruitment Status	Timing Administration	Malignancies Included	Clinical Trials Identifier
Idelalisib vs. Placebo	I	Recruiting	Maintenance post allo-HSCT	MCL, CLL, FL, DLBCL, B cell-tumors	NCT03151057
Anti-CD19-CAR-transduced T cell	I	Recruiting	Salvage post allo-HSCT	NHL, HL, B cell-tumors	NCT01087294
Ibrutinib	II	Ongoing, but not currently recruiting	Salvage post allo-HSCT	MCL, CLL, FL, HL, B cell-tumors	NCT02869633
Ibrutinib and Selinexor	I	Recruiting	Salvage post allo-HSCT *	MCL, DLBCL, CLL, SLL, PLL	NCT02303392

* Patients who have undergone autologous or allogeneic stem cell transplant = < 4 weeks prior to cycle 1 day 1 are excluded. Abbreviations: CLL (Chronic Lymphocytic Leukemia), SLL (Small Lymphocytic Lymphoma), PLL (Prolymphocytic Leukemia), FL (Follicular Lymphoma), MCL (Mantle Cell Lymphoma), DLBCL (Diffuse Large B-Cell Lymphoma).

**Table 3 cancers-13-00291-t003:** Comparison between CAR-T KTE-X19 therapy and allo-SCT.

Study	Approach	N° Patients	Median Age (Range)	N° Prior Lines (Range)	N° Refractory (%)	N° Prior Rituximab (Range)	N° Prior iBTK (Range)	ORR/CR	Median FU	1-yr NRM	1-yr PFS	1-yr OS
Hamadani et al.	MAC	74	54 (27–69)	3 (2–5)	37 (50)	11 (52)	0	?	35 mo	43%	31%	33%
Hamadani et al.	RIC/non -myeloablative	128	59 (42–75)	4 (1–5)	71 (55)	52 (80)	0	?	43 mo	38%	38%	46%
Cook et al.	RIC	70	52.2 (34.7–68.8)	2 (1–6)	12 (17%)	40 (64%)	0	51/48%	37 mo	18%	≈50%	≈75%
Le Gouill et al.	RIC	70	56 (33–67.5)	2 (1–5)	15 (21%)	?	0	94%/89% of eval. pts	24 mo	≈20%	?	≈60%
Wang et al.	CAR-T	68	65 (38–79)	3 (1–5)	27 (40)	68 (100)	68 (100)	85%/59%	12.3 mo	3%	61%	83%

[19,20,46,87].

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
