# Peer review of "Allogeneic Stem Cell Transplantation in Mantle Cell Lymphoma in the Era of New Drugs and CAR-T Cell Therapy"

_cancers, 2021, doi:10.3390/cancers13020291_

Round 1
Reviewer 1 Report
Excellent and comprehensive review on the role of allogeneic stem cell transplantation in patients with MCL in cluding the potential impact of the advent of CART cell constructs in this setting. Very well written and structured manuscript. No major comments from my side.
Author Response
Thank you for your comments regarding our manuscript, and for taking the time to provide valuable feedback.Reviewer 2 Report
The text is well written and organized.
It describes the therapeutic possibilities currently available for the treatment of MCL.
The transplant section describes the timing and the possible conditioning reported in the literature. Conservative/citoriductive pre and possibly post-transplantation therapy also reports the main results present in the literature. The section on CAR T is objectively still being under investigation.
I recommend the preparation of a summary table indicating the studies showing allogenic transplantation in the first or second line and the main outcomes and their toxicity.
Author Response
Thank you for your comments regarding our manuscript.
Following this right indication, we added at page 4 (par.4, Table 1) a summary table which compares studies on the use of allo-SCT after first line vs studies on the use of allo-SCT after subsequent lines of treatment.